# Recycling of Post-Consumer Polystyrene Packaging Waste into New Food Packaging Applications—Part 1: Direct Food Contact

**Frank Welle**

Fraunhofer Institute for Process Engineering and Packaging IVV, Giggenhauser Straße 35,
85354 Freising, Germany; frank.welle@ivv.fraunhofer.de

**Abstract:** The increase in plastic recycling is an essential pre-requisite for the transition to a circular economy. Polystyrene (PS) is a low diffusive polymer and therefore a promising candidate for recycling back into food contact similar to polyethylene terephthalate (PET). However, such a recycling of PS cups has been not established to date on a commercial scale. Even if recycling back into food contact is desired, the health of the consumer must not be at risk. As a consequence, recycling processes must go through a conservative assessment by relevant authorities. For PS, however, evaluation criteria are not published, which is a drawback for process developers. Within the study, post-consumer PS recyclates were evaluated in a similar way to existing evaluation criteria for PET and HDPE. For the recycling of post-consumer PS back into packages with direct contact with food, there are still some points open which cannot be answered conclusively today. Upon closer inspection, there appears to be enough information available to give a first indication as to whether recycling of post-consumer PS packaging materials back into direct food contact can be considered safe. The knowledge gaps in PS recycling were determined and discussed.

**Keywords:** high impact polystyrene; migration; yogurt cups; exposure evaluation





## 1. Introduction

In 2015, the European Commission published its first circular economy action plan, which identifies the increase in plastic recycling as an essential pre-requisite for the transition to a circular economy. The European (EU) Commission committed to address plastic recycling in a targeted way. As a consequence, the EU Commission therefore adopted a European strategy for plastics in a circular economy in 2018 [1]. On the other hand, the increase in post-consumer recycled plastics in food packaging must not endanger the health of the consumer. According to Regulation (EC) 282/2008 [2], recycled plastics in direct contact with food may be obtained only from processes which have a safety assessment of the European Food Safety Authority (EFSA) followed by an authorization by the EU Commission. That regulation was recently repealed and replaced by Regulation (EU) 2022/1616 [3]. The new regulation fundamentally changes the formal approval process of the EU Commission for recycled post-consumer plastics in food packaging materials, whilst not necessarily changing all the technical safety evaluation criteria developed by EFSA. Among many other things, it introduces the categories "suitable" and "novel" recycling technologies. Novel recycling technologies are those on which the European Commission has not yet taken a final decision, based on an opinion by EFSA. Most mechanical recycling technologies are considered "novel", except for polyethylene terephthalate (PET) technologies. Crucially, the new regulation currently foresees and requires authorizations only for each individual PET recycling process under the umbrella of the suitable PET recycling technology. Novel technologies, like mechanical PS recycling processes, must be notified. This notification, after assessment of the relevant authorities, allows the use of recycled plastic in food contact materials, but requires a monitoring and detailed reporting e.g., of any contaminants. Based on an assessment by EFSA, the EU Commission decide whether the recycling technologies will be classified as "suitable" technology.

Under the old, never fully implemented, regime of Regulation (EC) 282/2008, very strict requirements for the safety of recycled plastics have led to an industry practice to recycle only post-consumer PET into food packaging for direct food contact applications [4–6]. There are several reasons why only post-consumer PET bottles can be recycled into new packaging applications. PET collection systems, including curbside and deposit schemes, are well established all over Europe, which generates a huge input stream of post-consumer PET bottles. This led to innovative developments of PET super-clean recycling processes with very good cleaning efficiencies. In addition, PET is a low diffusive polymer, which means that the concentrations of substances absorbed into the PET bottle wall during the first life of the beverage bottle is low [7]. Such low input contamination levels together with high cleaning efficiencies of the super-clean recycling processes make PET the ideal packaging material for a circular economy [6–10]. Due to the low diffusivity of PET, and therefore the very low migrated amount of potential contaminants from PET into food, the use of post-consumer PET into new beverage bottles with up to 100% recyclate content is possible [10]. In contrast, the recycling of PET trays is very limited. This is because such PET trays are collected only in curbside collections (such as green dot systems) which are less controlled, and the trays contain often sealing layers, printed parts or adhesive labels, which are hard to remove during recycling. The collection of the post-consumer waste has therefore a strong influence on the quality of the post-consumer recyclates [11–13].

The EFSA evaluation criteria for post-consumer plastics in direct contact with food are very conservative, which leads to a high level of protection of the consumer and, as a consequence, a very restricted use of post-consumer polymers in direct contact with food [14]. Due to the conservative evaluation criteria of EFSA, high diffusive polymers such as HDPE are completely excluded from such a recycling back into food contact at the moment [15]. Polystyrene (PS) is another low diffusive polymer, and therefore a promising candidate for recycling similar to PET [16]. However, recycling of PS back into food contact has not been established to date. The reason for this might be the lack of well controlled and sorted input streams, but is also due to the fact that super-clean recycling processes for PS are not available on an industrial scale. In addition, PS is used in two forms for food packaging: opaque high-impact polystyrene (HIPS) as well as crystal-clear general-purpose polystyrene (GPPS) [17]. However, the diffusion behavior of HIPS and GPPS, and therefore their migration potential, are similar [18]. Therefore, both forms can be collected and recycled together.

A general requirement for a recycling of packaging materials is that the recycling processes reduces possible post-consumer contamination from the recycled polymers to levels which do not endanger the consumer. Critical contaminants in post-consumer polymers might be chemicals from possible misuse of packaging containers, contaminants from non-food applications such as non-authorized additives as well as degradation products generated during recycling [4]. Other contamination, such as microbiological or viral contamination, can be excluded because of the high temperatures used to process the polymer [4]. To evaluate the risk for the consumer, EFSA defined an evaluation procedure, which is based on the following parts:

- Concentration of potential contaminants in post-consumer polymers;
- Cleaning efficiencies of the super-clean recycling processes;
- Exposure scenario of the consumer.

The concentration of potential contaminants in the post-consumer polymers (input concentrations) is the average concentration of substances introduced into the material after use of the package or substances from potential misuse, which should be determined on a statistical basis. This parameter is typically determined within huge post-consumer recyclate screening campaigns with thousands of samples. The huge number of samples is necessary in order to determine the low incidence of misuse. For example, the incidence of abuse PET bottles was determined to be 0.03–0.04% [5,14,19,20], which means that 3 or 4 PET bottles per 10,000 bottles show hints of chemicals from their misuse. The number of samples in such a screening campaign need to be in the range of 10,000 bottles; otherwise,

the incidence of abuse cannot be determined. On the other hand, the concentrations of misused chemicals in the input materials for the recycling process are relevant. The concentration in an individual misused bottle will be high, but due to the high dilution with non-contaminated bottles, the average concentration in the input stream is low. The average concentration is assumed as input concentration level of hazardous chemicals in the recycling process. EFSA assumed that all of these chemicals are genotoxic compounds [4], which, while being precautionary, is likely to be highly over-conservative [14].

The cleaning efficiency of the super-clean recycling process is understood as the potential of the recycling process to eliminate chemical contaminants from the first use or from misuse of the packaging material. Cleaning efficiencies of super-clean recycling processes are typically determined by so-called "challenge tests". Within a challenge test, the post-consumer recyclates are artificially contaminated with model contaminants (surrogates). These deliberately introduced surrogates are spiked in worst-case concentrations to the post-consumer recyclates. The concentrations, which are applied in a challenge test, are typically a factor of 100 higher as the worst-case initial concentration in the post-consumer input material. After a storage time at higher temperatures, typically 7 days at 50 °C, the contaminated material is then added to the recycling process under investigation without dilution with non-contaminated material. Samples are drawn during super-clean recycling and analyzed for their residual concentration of the applied surrogates. The difference in the concentrations of the individual surrogates between the artificially contaminated input materials and the output materials after the recycling process is the cleaning efficiency of the super-clean process. Cleaning efficiency data of PET and HDPE super-clean recycling processes are available [21–25].

The exposure scenario of the consumer considers the intake of the above-mentioned contaminants by the consumer. The exposure of the consumer is calculated from the daily food consumption of the relevant consumer groups (e.g., infants, toddlers or adults) and the migration of potential contaminants from the recyclate-containing packaging material into food. The migrated amount is typically not experimentally determined, but predicted by use of diffusion modelling. For example, EFSA is using the so-called $A_P$ prediction model for the evaluation of the migrated amount of the surrogates into food. This $A_P$ model over-estimates the real migration into food [26]. The extent of over-estimation (over-estimation factor) is, however, not known. This represents a significant drawback of the evaluation of the safety of post-consumer recyclates in direct food contact [14].

For the circular economy to deliver on the scale that has been promised, plastic recycling has to be extended beyond simply PET bottles. In this regard, PS is perhaps the next promising candidate worthy of consideration, but there are hurdles that will need to be overcome before this can be realized. Recovered PS from post-consumer sources is available, and recycled PS cups are intended for reuse into new food packaging materials. However, no mechanical recycling processes for post-consumer PS in direct contact with food have been established on industrial scale to date. Additionally, the food law compliance of PS super-cleaning recycling processes has not been evaluated to date. As a consequence, the evaluation criteria for the safe recycling of post-consumer PS in direct food contact have not been established. This is a major drawback and inhibits investment in this area.

The aim of this study is therefore to determine the state-of-the-art of PS recycling for the re-use of the PS recyclates in direct contact with food. Post-consumer PS recyclates were evaluated in a similar way to the existing EFSA evaluation criteria for PET and HDPE, and knowledge gaps were determined. In addition, experimental migration kinetics from an artificially spiked HIPS sheet were measured in order to investigate the amount of contaminants which might migrate from recycled materials into packed food. The migration data were used for the risk assessment of substances migrating from recyclate-containing PS yogurt cups, especially for the determination of the over-estimation factors of the applied migration prediction model.

## 2. Results

### 2.1. State-of-the-Art in Collection, Sorting and Recycling of Post-Consumer PS

The sorting and conventional recycling of post-consumer PS typically contains the following steps [27]: The sorting of post-consumer waste starts from waste bales, which consist of articles from household waste from green dot systems and curbside collections. Due to a first sorting step at municipalities, these waste bales contain mainly PS articles. At the recycling facility, the PS waste is fed onto the transport belt, where metals and non-PS-containing articles (such as PET, polyolefins, etc.) are sorted out by standard near infrared (NIR) technology. Subsequently, the obtained PS stream is further sorted, and any non-food packaging articles are removed by either object recognition-based artificial intelligence or humans. This sorting step removes non-food PS materials such as CDrom packaging, flowerpots, clothes hangers, cosmetic packaging, etc. The final stream of the sorting facility is rigid PS that was in use for food packaging in the first life. In addition, expanded PS is sorted out from the input stream. So, only rigid and semi-rigid PS are used as input material for the recycling process.

This sorted PS stream is then sent to the shredder, which reduces the size of the articles to flakes of about 1 cm$^2$. The following washing step comprises typically a hot pre-washing and an intense washing step using 1% caustic soda at a temperature of >80 °C, followed by rinsing and a density separation of polymers other than PS. After the washing step, the flakes are mechanically and thermally dried in order to bring the water level down to virtually 0%. The washed and dried PS flakes are then introduced into color sorting as well as polymer sorting process using near-infrared (NIR) technology, resulting in a white and colored fraction of PS flakes, each of 99.8% purity.

For the re-use of the recyclates in direct food contact, the PS flakes need to be further deep-cleansed by use of super-clean recycling processes. To the knowledge of the author, no cleaning efficiencies of such PS super-clean recycling processes are published to date. However, three petitions were submitted to EFSA for recycling of post-consumer PS into new packaging applications according to Regulation (EC) 282/2008, but were terminated as a consequence of the new Recycling Regulation (EU) 2022/1616 [3]. As explained above, PS recycling now has to be notified as "novel" technologies according to the Regulation (EU) 2022/1616. Due to confidentiality of the EFSA evaluation process, it is not possible to obtain information on which recycling process steps had been used in the above-mentioned EFSA dossiers. However, it is most likely that the PS super-clean recycling processes rely on similar recycling processes that are used for PET, such as re-extrusion and thermal decontamination processes in solid or liquid state. The principles of PET super-clean processes are described in review articles [14,28]. It is perhaps speculative, but one might expect that the decontamination efficiency of PS super-clean recycling processes will be lower than that for PET. This could be anticipated since PS has a significantly lower melting point and, most likely, it will be possible to apply a lower decontamination temperature.

### 2.2. Input Contamination Levels

With regard to the input contamination in recycled flakes (rPS) before super-clean processes, scientific literature studies on misuse or contamination levels from substances from the first use of the package (post-consumer substances) are rare. Only a limited amount of post-consumer PS samples could be analyzed regarding post-consumer substances. In these samples, post-consumer substances such as limonene were found in the range of 1 mg/kg, and the highest concentration of limonene was determined to 12 mg/kg [29]. It should be noted that this study was conducted about 25 years ago. The PS recycling processes and the input streams have been strongly developed in the last decades. Newer studies identified substances in rPS, but quantitative information is not available [30,31]. However, quantitative data on post-consumer substances is a pre-requisite for the safety evaluation of post-consumer PS.

Regarding substances from misuse, the number of samples is currently too low for a clear conclusion. However, hints for misused substances were not found in any investigated

samples to date. This is perhaps to be expected since PS containers, in contrast to PET bottles, typically cannot be re-sealed, and PS as polymer is less chemically resistant, rendering it less suitable for storage of chemicals. If the true incidence of misuse in post-consumer PS packaging is much lower than that for PET, then a considerably larger sample size compared to PET would be needed to achieve the same statistical power of the incidence of misused PS packaging [14]. In the absence of such a determination for PS, the use of the misuse rate of PET bottles (3–4 per 10,000) would appear to be a conservative approach.

### 2.3. Experimental Migration Kinetics

In order to obtain an impression about the migrated amount of potential post-consumer substances in rPS systematic migration, tests were performed. Due to the fact that post-consumer substances are only determined in low concentrations [29], HIPS sheets artificially spiked in high concentrations with surrogates typically used in challenge tests were prepared for the migration tests. Spiking was performed during sheet production so that the HIPS sheet was homogenously contaminated with the surrogates. The migration kinetics were determined at temperatures of 5 °C, 20 °C, 40 °C and 60 °C for an overall time of 10 days for each migration kinetic. For all investigated migrants, kinetic points were determined every 40 min. Therefore, a total of 360 kinetic points were obtained for each surrogate and each temperature. The migration curves are given in Figure 1. The results of the migration kinetics after storage for 10 days at 5 °C, 20 °C, 40 °C and 60 °C (in µg/cm²) are given in Table 1. As expected, the migrated amount of the surrogates shows a strong increase with increasing temperature. The specific migration of the different substances at 60 °C is roughly a factor of 100 higher than that observed at 5 °C.

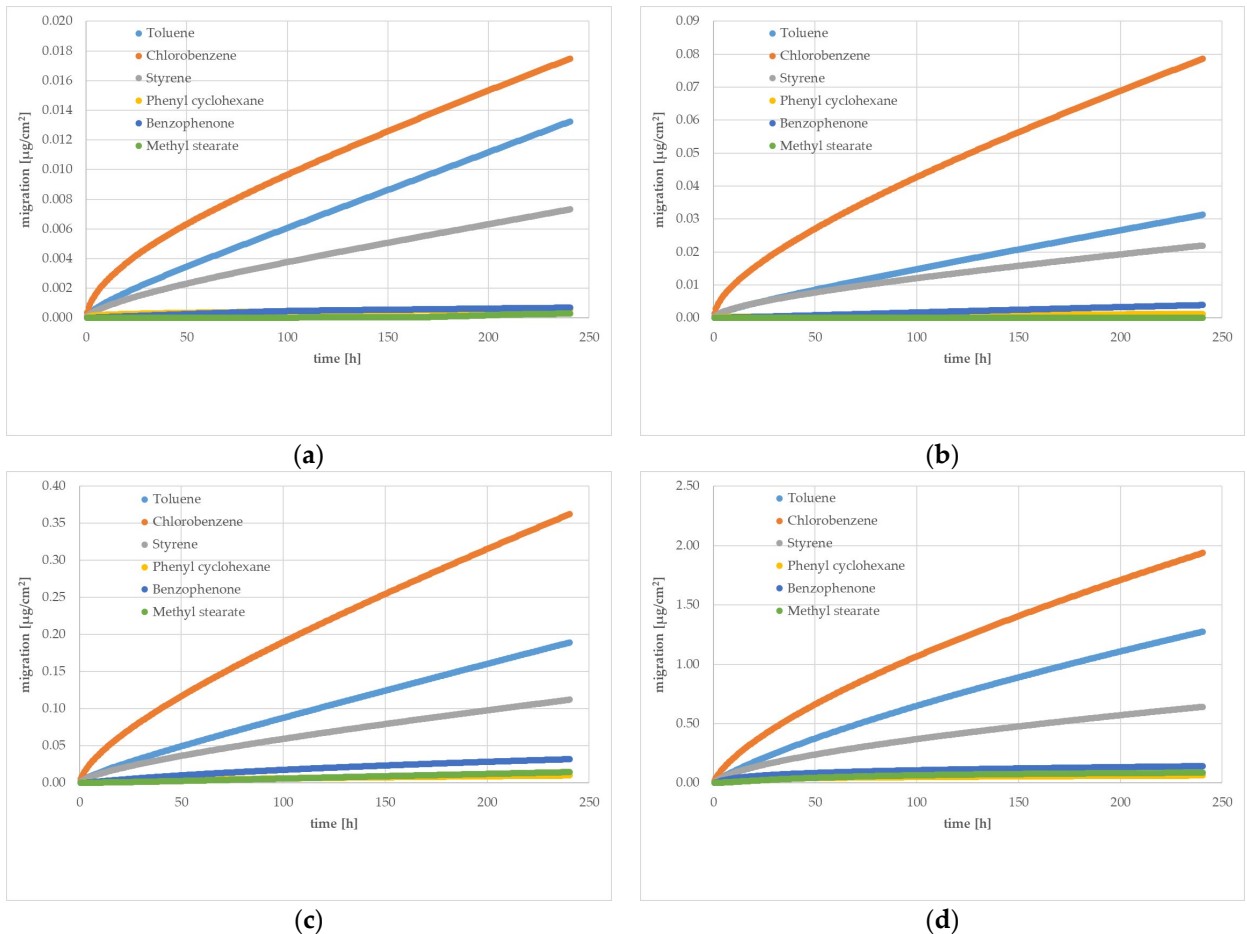

**Figure 1.** Experimentally determined gas phase migration kinetics of from HIPS at 5 °C (**a**), 20 °C (**b**), 40 °C (**c**) and 60 °C (**d**). Spiking levels for the different substances are indicated in Section 2.4.

**Table 1.** Experimentally determined migrated amount into the gas phase from HIPS after storage for 10 days at 5 °C, 20 °C, 40 °C and 60 °C.

| Temperature | Migrated Amount [$\mu g/cm^2$] | | | | | |
|---|---|---|---|---|---|---|
| | Toluene | Chlorobenzene | Styrene [1] | Phenyl Cyclohexane | Benzophenone | Methyl Stearate |
| 5 °C | 0.0132 | 0.0175 | 0.00732 | 0.000581 | 0.000691 | <0.0001 [2] |
| 20 °C | 0.0313 | 0.0786 | 0.0220 | 0.00122 | 0.00391 | <0.0001 [2] |
| 40 °C | 0.189 | 0.363 | 0.112 | 0.0101 | 0.0321 | 0.0146 |
| 60 °C | 1.27 | 1.94 | 0.642 | 0.0629 | 0.141 | 0.0870 |

[1] Residual monomer, not artificially added, [2] below the experimental detection limit of 0.0001 $\mu g/cm^2$.

### 2.4. Exposure Calculations According to the EFSA Procedure

The evaluation of the consumer safety of post-consumer recyclate was based on the Threshold of Toxicological Concern (TTC) Concept [32,33]. The TTC approach has also been used by EFSA for the safety evaluation of post-consumer PET [4,5] and HDPE [15] recyclates. In principle, not all post-consumer substances can be analyzed in every PS flake or pellet sample and, in addition, some of the post-consumer substances cannot be identified. Therefore, the consumer safety evaluation of post-consumer evaluation is based on the assumption that non-identified substances are genotoxic. According to the TTC concept, an exposure of 0.0025 μg of a substance per kg body weight per day is not critical to human health. The exposure-related amount (or limit) of a substance can be calculated with the body weight and the daily consumption. This limit (in μg per kg food) can be considered as the maximum admissible value for the migrated quantity of a post-consumer substance from the recyclate-containing PS cup or tray into food at the end of shelf life.

As mentioned previously, to date EFSA has not published evaluation criteria for post-consumer PS in direct food contact. In the absence of EFSA criteria, possible food contact applications of PS were identified and evaluated (Table 2). The applications include yogurt cups, trays for meat, fish, cheese and fruit as well as drinking cups. Consumer consumption data for milk products and meat and fish are derived from the EFSA food consumption database [34]. The consumption of fruit and vegetables was estimated. The consumption data for drinks were derived from the EFSA evaluation of PET recycling for mineral water bottles [5]. It should be noted that the food consumption database does not consider the packaging materials. The above-mentioned consumption data for food consider all kinds of packaging materials, not only PS, because yogurt for example is also packed in polypropylene (PP). Meat is also packed in PET trays. Considering PS as the only packaging material for the above-mentioned foods can be therefore considered as the worst case. The surface volume ratio was assumed to 600 $cm^2$/1000 mL for all applications, which was also used by EFSA for PET mineral water bottle evaluation [5]. For cups and trays, this surface volume ratio can be also considered worst case, because only the lower part of the packaging is made out of PS. EFSA assumed for PET that the $A_P$ prediction model over-estimates the migration by a factor of 5 as a minimum [5]. In order to be comparable to the EFSA evaluation for PET beverage bottles, an over-estimative factor (OEF) of 5 was also used in Table 2.

**Table 2.** Migration scenario for calculations for toddlers with 10 kg body weight. Values for food consumption, maximum storage temperatures, admissible exposure and values for maximum migrated quantity are indicated.

| Application | Food Consumption per Day | Maximum Storage Time and Temperature | Exposure (Maximum Migration Value) | Maximum Migration with Overestimation Factor 5 |
|---|---|---|---|---|
| Cold-filled yogurt | 250 g | 40 d at 6 °C | 0.1 μg/kg | 0.5 μg/kg |
| Hot-filled yogurt | 250 g | 2 h 70 °C followed by 40 d at 6 °C | 0.1 μg/kg | 0.5 μg/kg |
| Trays for meat, fish or cheese | 150 g | 30 d at 6 °C | 0.167 μg/kg | 0.835 μg/kg |
| Trays for food and vegetables | 500 g | 30 d at 25 °C | 0.05 μg/kg | 0.25 μg/kg |
| Cups for cold drinks | 750 mL | 1 d at 25 °C | 0.033 μg/kg | 0.167 μg/kg |
| Cups for hot drinks | 750 mL | 2 h at 70 °C | 0.033 μg/kg | 0.167 μg/kg |

### 3. Discussion

#### 3.1. Input Contamination Levels

Data on the input contamination levels are rare in the scientific literature, and even if they are present, they are not up to date regarding the applied sorting and recycling steps. Statistical data for the misuse of PS container for storage of (solid) household chemicals are not available to date.

Looking into the data available for PET, 0.03% to 0.04% of the recycled PET beverage bottles show hints of misuse [5,14,19,20], which is a low incidence. Toluene has been identified as an example of the sort of substances which are most likely filled into these misused PET bottles [19]. In terms of consumer behavior, PET bottles are much more suitable for storage of liquids, because the bottles can be re-sealed with a closure. PS cups or trays cannot be re-sealed, and are therefore not suitable for storage of liquid chemicals. In addition, solvents such as toluene dissolve PS and destroy the container. Therefore, the incidence for misuse of PS cups or trays for storage of hazardous chemicals is most likely much lower than that for PET. This lower incidence will make it very difficult to determine the incidence of misuse experimentally, since large numbers of post-consumer PS flakes samples need to be analyzed. If one assumed a factor of 10 lower incidence for PS compared to PET, the same factor of 10 higher number of post-consumer PS samples (100,000 PS cups or trays) will need to be analyzed in order to obtain the same statistical basis regarding misuse of PS as for PET. Such an experimental study will be expensive and time consuming.

As mentioned above, PS cups cannot be resealed; therefore, they would be unsuitable for storing liquids. If only solid chemicals were filled into PS cups, then, since these compounds are high molecular substances with very low diffusion rates, their sorption into PS will be very low, and contamination would only be on the outer layer. A low incidence of misuse and a low sorption rate will result in (very) low initial concentrations of potential contaminants in washed PS flakes, which is the input stream of super-clean recycling processes. Again, experimental data on a statistical basis are not available in the scientific literature to date. Therefore, as a pragmatic approach, the same input concentration as for PET of 3 mg/kg might be suitable for the safety evaluation of PS recyclates. According to the above discussion, the concentration should be significantly lower than 3 mg/kg, and the value should only be considered as a basis for a first safety assessment of PS recyclates.

#### 3.2. Non-Food Input Levels

In case of PET super-clean recycling processes, EFSA concluded that non-food PET containers should not be intentionally used as feedstock input, and that the non-food fraction should not exceed 5% of the input material [5]. Due to the comparable low diffusion behavior of PET and PS, the 5% value might be also applicable for PS. This value should not be exceeded in the PS recycling process by sorting steps which include object recognition combined with artificial intelligence and humans.

It should be noted that non-food PS articles are typically manufactured from food-grade PS, and at least all non-food objects were produced from virgin PS polymers. However, there are applications in construction where PS polymer producers sell non-food PS, which has a significantly higher residual monomer (styrene) concentration. In contrast, the additives used in PS intended for construction are the same as those used in food contact materials [27]. Provided that expanded PS objects are separated in the sorting process and expanded PS objects do not enter the PS recycling stream considered here (see Section 2.1), contamination of the PS recyclates by non-food PS is unlikely.

#### 3.3. Over-Estimative Factors (OEF) for the Applied Surrogates

EFSA is applying diffusion models for the conversion of exposure values into migration limits and maximum concentrations of a contaminating substance in the packaging. The latter are compared to the results from the challenge test. The $A_P$ prediction model had been used by EFSA for the evaluation of mechanical PET and HDPE recycling pro-

cesses [5,15]. Most likely, EFSA would apply the $A_P$ model for the evaluation of PS recycling processes. The $A_P$ prediction model is conservative, which means that the predicted migrated amount is in any case higher than the real migration into food. This is a pre-requisite of the Regulation 10/2011 where it is stated that prediction models should be "constructed such as to over-estimate the real migration" [35]. A major drawback of the conservative character of the $A_P$ prediction model is that OEF for each migrant is not known. In order to address this over-estimative tendency of the applied $A_P$ model, EFSA uses an OEF of 5 (PET) [5] and 2 (HDPE) [15] for all applied surrogates used in the challenge test.

Within this study, the OEFs of the $A_P$ model were derived for the applied surrogates from the comparison between the experimental migration kinetics data (Table 1) and the predicted migrated amount (Table 3). As expected, the OEFs are depending on the molecular weight of the substance and the storage temperature. For low molecular weight molecules such as toluene and chlorobenzene, the OEFs for the $A_P$ model are smaller than for high molecular substances such as phenyl cyclohexane and benzophenone. The OEF also decreases at higher temperatures due to the fact that the activation energy in the $A_P$ model is fixed for all substances to 87 kJ/mol [26], whereas the true activation energies of diffusion vary and in most cases are higher than 87 kJ/mol, especially for high molecular weight substances [18].

**Table 3.** Predicted migrated amount of surrogates from HIPS with the storage conditions 10 days at 5 °C, 20 °C, 40 °C and 60 °C (K = 1, density 1.04 g/cm$^3$, layer thickness 300 μm) and over-estimative factors from experimental migration kinetics (Table 1) compared to the modelling parameters for HIPS ($A_P'$ = 1.0 and τ = 0 K).

| Temperature | Predicted Migrated Amount [μg/cm$^2$] (over-Estimative Factors Compared to Experimental Migration) | | | | | |
|---|---|---|---|---|---|---|
| | Toluene | Chlorobenzene | Styrene [1] | Phenyl Cyclohexane | Benzophenone | Methyl Stearate |
| 5 °C | 0.275 (20.8) | 0.249 (14.2) | 0.118 (16.1) | 0.265 (456) | 0.187 (271) | 0.113 (>1130) |
| 20 °C | 0.717 (22.9) | 0.651 (8.31) | 0.309 (14.0) | 0.693 (568) | 0.488 (125) | 0.294 (>2940) |
| 40 °C | 2.24 (11.8) | 2.03 (5.61) | 0.964 (8.60) | 2.16 (213) | 1.52 (47.5) | 0.917 (64.0) |
| 60 °C | 6.08 (4.77) | 5.52 (2.86) | 2.62 (4.08) | 5.88 (93.2) | 4.14 (29.3) | 2.50 (28.6) |

[1] Residual monomer, not artificially added.

It is important to note that the migration kinetics into the gas phase cannot faithfully reflect partitioning between the polymer and the gas phase. Therefore, the partition coefficient is virtually K = 1, which was also used for the prediction of the migrated amount, either by EFSA [5,15] or in this study (see Section 4.4).

The data from the migration kinetics show that the lowest OEFs are found for chlorobenzene (2.86 to 14.2). So, from a migration prediction point of view, chlorobenzene is the most demanding substance of the challenge test surrogated cocktail. From a cleaning efficiency point of view, however, chlorobenzene is not the most demanding surrogate. The highly volatile surrogates (toluene and chlorobenzene) are most probably efficiently removed in the decontamination process. The low volatile surrogates phenyl cyclohexane, benzophenone and methyl stearate are much more demanding. These low volatile substances will have much higher residual concentrations in the final recyclate compared to the volatile substances. The low volatile surrogates, however, have much higher OEFs up to 568 (phenyl cyclohexane), 271 (benzophenone) and >2940 (methyl stearate) (Table 3). Therefore, a fixed OEF is significantly influencing the overall evaluation of PS recyclates in direct contact with food. Low volatile substances with high residual concentrations in the challenge test will be predicted with the highest OEF by the $A_P$ model. As a consequence, either realistic prediction models or realistic OEFs should be applied in the safety evaluation of PS recycling processes instead of a fixed OEF of 5. When including an OEF in the diffusion modelling approach, it should be made certain that the exposure is realistic or only slightly over-estimative. The degree of over-estimation depends on the diffusion behavior of the migrants in the PS polymer. Therefore, individual OEFs seem to be much more suitable

than a fixed OEF for all applied surrogates. These individual OEFs are important because otherwise the safety evaluation criteria for the individual surrogates will be different. For example, phenyl cyclohexane has an OEF of 568 at 20 °C and 456 at 5 °C. The migrated amount into food is therefore by the same factor lower than the predicted amount. A safety factor of approximately 500 would appear to be excessively conservative, especially bearing in mind that the initial concentration, the cleaning efficiency and the exposure to the consumer were also established under worst-case considerations.

### 3.4. Minimum Cleaning Efficiencies of Super-Clean Recycling Processes

In Table 2, the maximum admissible exposure of the consumer for six applications of post-consumer recyclates are summarized. From these data, the migration threshold limit was derived by applying the conservative $A_P$ model and a fixed OEF of 5. The downsides of using a fixed, conservative OEF of 5, as discussed above, is justified by the need for a value for the calculation of the minimum cleaning efficiency of potential super-clean recycling processes of PS, and EFSA used this value for the evaluation of PET. The minimum cleaning efficiencies can be derived from the worst-case input contamination level and the maximum migration for the rPS applications given in Table 2.

The input contamination level of 3 mg/kg is assumed as the worst-case initial concentration in the cup or tray (see Section 3.1). Based on this data, the maximum migrated amount into packed food can be calculated in a way that is very similar to the EFSA approach. Subsequently, the corresponding concentration in the packaging material can be calculated. As an example, the migrated amount of the surrogate toluene (molecular weight: 92 g/mol) with a starting concentration of 3 mg/kg was calculated by use of the $A_P$ prediction model to 1.38 µg/kg in yogurt after storage for 40 days at 6 °C (storage conditions of cold-filled yogurt). From the exposure given in Table 2, maximum migration, however, is only 0.5 µg/kg for this application. The calculated migrated amount with 3 mg/kg as concentration of a substance in the packaging material is a factor of 2.76 too high. Reducing the concentration in the packaging material to 1.08 mg/kg leads to a (predicted) migration value of 0.5 µg/kg. Consequently, the super-clean recycling process has to reduce the concentration of the surrogate toluene from 3 mg/kg to 1.08 mg/kg, which is a cleaning efficiency of 64.0%. Comparable calculations were performed for all surrogates, maximum migration values and storage conditions. The calculated minimum cleaning efficiencies of the recycling process for the applications and food consumption data from Table 4 are given in Figure 2.

**Table 4.** Minimum cleaning efficiencies for the super-clean recycling process for the applications given in Table 2 (over-estimative for the calculation is factor of 5 as applied by EFSA for low diffusive polymers).

| Application | Minimum Cleaning Efficiency | | | | |
|---|---|---|---|---|---|
| | **Toluene** | **Chlorobenzene** | **Phenyl Cyclohexane** | **Benzophenone** | **Methyl Stearate** |
| Cold-filled yogurt | 64.0% | 57.2% | 39.9% | 30.4% | 0% |
| Hot-filled yogurt | 80.0% | 75.1% | 66.5% | 61.3% | 24.5% |
| Trays for meat, fish or cheese | 30.6% | 17.6% | 0% | 0% | 0% |
| Trays for food and vegetables | 93.7% | 92.5% | 89.4% | 87.8% | 76.2% |
| Cups for cold drinks | 76.9% | 72.6% | 61.5% | 55.4% | 13.0% |
| Cups for hot drinks | 91.9% | 90.4% | 86.5% | 84.5% | 69.7% |

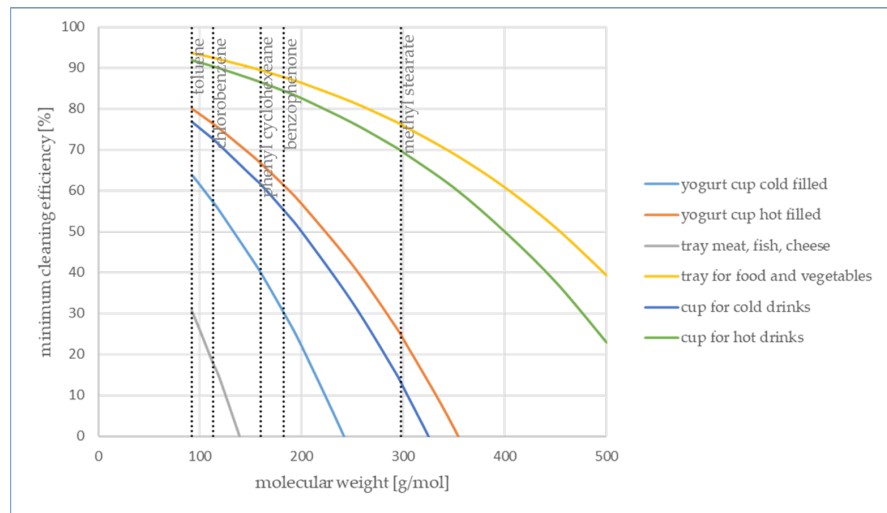

**Figure 2.** Minimum cleaning efficiencies for the applications given in Table 4 calculated with an OEF of 5 of the $A_P{}'$ Model.

As shown in Figure 2, the trays for food and vegetables have the highest requirements on the cleaning efficiency of the super-clean process. This is due to the long storage time of 30 days at 25 °C. On the other hand, in case of vegetables, the food is washed and often peeled so that the migrated substances are most probably removed. The exposure to the consumer is therefore much lower. In case of cold-filled yogurt, the cleaning efficiency requirements on the super-clean recycling process are much lower compared to the exposure scenario of trays for food and vegetables. For cold-filled yogurts, the cleaning efficiency for the volatiles substances toluene and chlorobenzene are 64.0% and 57.2%, respectively. The low volatile substance methyl stearate does not need to be removed during super-cleaning. This means that the concentration of methyl stearate is allowed to be higher than 3 mg/kg, which is the initial concentration assumed for the calculations. No cleaning efficiency is required in general for substances with a molecular weight >250 g/mol under the applied assumptions for the safety evaluations of cold-filled yogurts.

As mentioned above, the cleaning efficiency calculations were performed assuming an OEF of the applied $A_P$ model of 5. In case of cold-filled yogurts, the realistic OEFs are available for 5 °C (Table 3), which is close to 6 °C used for the safety evaluation. It is interesting to note that for cold-filled yogurt no cleaning efficiency is necessary for all surrogates if realistic OEFs are applied instead of a factor 5. This means that the application of cold-filled yogurt can be considered as safe without any super-clean process being applied. However, the extrusion of PS flakes is in any case part of the recycling and/or packaging manufacturing process. The cleaning efficiency of this re-extrusion process due to high temperatures will add an additional margin of safety into the PS recycling process. The same conclusions can be drawn for trays for meat fish or cheese because they are stored at 6 °C for 30 days.

For all other applications given in Table 2, it is more difficult to apply a realistic modelling approach because the OEFs were determined at 20 °C, 40 °C and 60 °C instead of 25 °C and 70 °C. In case of hot-filled yogurt, two different sets of OEFs must be applied because two temperatures (70 °C and 6 °C) are applied in the calculations. However, the OEFs up to 40 °C given in Table 3 are in any case higher than the applied factor of 5. Only at 60 °C for the high volatile substance is the OEF slightly below 5. In case of hot-filled yogurt, the OEFs at 5 °C will most likely overcompensate when realistic modelling is applied.

## 4. Materials and Methods

### 4.1. Manufacturing of HIPS Sheet Spiked with Model Compounds

A spiked sheet was prepared from virgin HIPS polymer for the migration kinetics into the gas phase. Due to the fact that migration kinetic were determined from the

spiked samples, virgin HIPS was used instead of post-consumer recycled PS. This is to minimize interference with substances from the first use. In addition, model compounds (surrogates) were used instead of real contaminants because polystyrene samples with true contaminants in the molecular weight range of interest are rarely available. Acetone, ethyl acetate, toluene, chlorobenzene, phenyl cyclohexane, benzophenone and methyl stearate were used as model substances (Sigma-Aldrich, Saint Louis, MO, USA). The surrogates were introduced into the HIPS sheet during extrusion of the sheet in order to obtain a homogeneous distribution of the surrogates in the sheet. The thickness of the sheet was determined to 400 ± 30 μm.

*4.2. Quantification of Spiking Levels in the Polystyrene Sheet*

The concentrations of the applied surrogates were determined quantitatively in the HIPS sheets by use of an extraction procedure with dichloromethane followed by gas chromatographic analysis of the extracts. The procedure was applied as follows: 1.0 g of the HIPS material was extracted with 10 mL dichloromethane at 40 °C for 3 days. The extracts were decanted from the polymer and analyzed by gas chromatography with flame ionization detection (GC-FID): Column: DB 1—20 m—0.18 mm i.d.—0.18 μm film thickness, temperature program: 50 °C (2 min), followed by heating at 10 °C/min to 340 °C (15 min), pre-pressure: 50 kPa hydrogen, split: 10 mL/min. Calibration was achieved by standard addition of the model compounds. Additionally, *tert*-Butylhydroxyanisole (BHA, CAS No. 8003-24-5) (Sigma-Aldrich, Saint Louis, MO, USA) and Tinuvin 234 (CAS No. 70321-86-7) (Sigma-Aldrich, Saint Louis, MO, USA) were used as internal standards. The concentrations of the surrogates in the investigated polystyrene sheets are summarized in Table 5. After the first extraction, the sheets were extracted again according to the abovementioned procedure. The concentrations in the second extract were below 5% of the concentrations in the first extracts, which indicates that the first extraction was exhaustive.

**Table 5.** Experimentally determined concentrations of model compounds in the spiked HIPS sheet.

| Surrogate | Spiked Concentration [mg/kg] |
| --- | --- |
| toluene | 763 ± 11 |
| chlorobenzene | 821 ± 13 |
| styrene [1] | 363 ± 6 |
| phenyl cyclohexane | 1229 ± 33 |
| benzophenone | 1002 ± 27 |
| methyl stearate | 1179 ± 34 |

[1] Residual monomer, not artificially added.

*4.3. Migration Kinetics into the Gas Phase*

Quantities that migrated into the gas phase of the spiked model compounds were determined according to literature. Some [36,37] used an automated method that involved placing sheet samples of 15.6 cm diameter (area 191 cm$^2$) in a migration cell. Migration into the gas phase was chosen because migration into food simulants significantly swells the PS matrix, which increases the migration. Real food, however, does not significantly swell the PS matrix [38]. The migration cell with the sheet sample was heated up to the measuring temperature. The surrogates, which are migrated out of the PS sheets, were purged out of the extraction cell by a helium stream of 20 mL/min. The surrogates were trapped (Carbopack B 20 mm, Supelco) at a trap temperature of −46 °C. Every 40 min, the loaded trap was completely desorbed (heating up to 300 °C within about 10 s) and transferred into the connected gas chromatograph (GC) (Thermo Scientific, Langenselbold, Germany). The surrogates were separated and quantified during the GC run. During running of the gas chromatograph, a new trapping cycle started. Calibration was achieved by injection of undiluted standard solutions of the surrogates into the migration cell. Gas chromatograph: Column: Rxi624, length: 30 m, inner diameter: 0.32 mm, film thickness

1.8 µm. Temperature program: 40 °C (2 min), rate 20 °C/min, 270 °C (8 min), pressure 70 kPa helium, detector temperature: 280 °C.

### 4.4. Diffusion Modelling

Diffusion modelling was performed using the AKTS SML software version 4.54 (AKTS AG, Siders, Switzerland). The program uses finite element analysis [39]. The modelling conditions are the same as given in the EFSA evaluations of mechanically recycled polymers [5,15]. All calculations were based on the modelling parameters for HIPS ($A_P' = 1.0$ and $\tau = 0$ K) [26]. As side conditions, a density of 1.04 g/cm$^3$ and high solubility for the surrogates (partition coefficient between polymer and food $K_{P,F} = 1$) and a recyclate content of 100% in the food contact article was used. The surface volume ratio was 600 cm$^2$/1000 mL for all applications and calculations. The wall thickness in the calculations was 300 µm, which is the worst case scenario for most of the packaging. It should be noted that due to the low diffusivity of HIPS, the wall thickness has no influence on the (calculated) migration result.

### 5. Conclusions

Recycling of post-consumer PS packaging waste back into new packaging has not been established in Europe on an industrial scale to date. The contamination levels in collected and washed PS flakes are also not known. As a logical consequence, the requirements regarding the cleaning efficiencies of the PS super-clean recycling processes are not clearly defined by relevant authorities. For the recycling of post-consumer PS into new packaging application with direct contact with food, there are still some points open which cannot be answered conclusively today. At first glance, it seems that without recycling processes in place, there will be neither data available on the cleaning efficiencies and input contamination levels nor evaluation criteria by the relevant authorities. As explained above, the new regulation aims to address this by allowing the use of plastic recycled with "novel" technologies in food contact materials and articles till the EU Commission, based on an assessment by EFSA that decides whether the PS recycling processes are "suitable" under certain conditions, which include monitoring and reporting. However, upon closer inspection, there appears to be enough information available to give a first indication as to whether recycling of post-consumer PS packaging materials back into direct food contact can be considered as safe.

The published EFSA opinions for the safety evaluation of post-consumer PET and HDPE recyclates in direct contact with food shows the evaluation principles which will most probably applied for post-consumer PS. A concentration of 3 mg/kg as the maximum concentration in post-consumer washed flakes was assumed in this study, following EFSAs evaluation of post-consumer PET. The cleaning efficiencies of super-clean recycling processes are also not available from the scientific literature. As a replacement for this lack of information, the minimum cleaning efficiencies were calculated in accordance to EFSAs evaluation principles (Table 4). These cleaning efficiencies can be considered as the starting point for the development of super-clean PS processes. The exposure of the consumer is estimated for possible applications for PS recyclates, which is typically performed by diffusion modelling.

The most favorable applications for post-consumer recyclates are foods for refrigerated storage conditions such as cold-filled yogurts or meat, fish or cheese. The low temperatures minimizes the migrated amount of post-consumer substances into packed food. In contrast to refrigerated storage condition, the migrated amount is much higher under hot-filled conditions. These applications necessitate the use of much higher cleaning efficiencies within the super-clean processes. However, the calculated cleaning efficiencies should be manageable by use of super-clean recycling technologies developed, due to similar low diffusivity of PS compared to PET.

As an overall conclusion, based on the data presented in this study and the evaluation therein, it can be concluded that the recycling of post-consumer PS recyclates can be



considered as safe, even if some knowledge gaps still exist. However, it should be noted that the eventual assessment of recycling technologies and, where applicable, processes is in the hands of EFSA. One of the most important factors in this evaluation of post-consumer PS recyclates, or in general for all recyclates in direct contact with food, is the migrated amount of post-consumer substances into food. By comparison of experimental migration values with predicted data, it could be shown that the applied $A_P$ prediction model is highly over-estimating the real migrated amount, especially for high molecular weight substances. EFSA considered this over-estimation by applying over-estimation factors (OEFs), but only in a general way. The findings of this study show that the use of a single OEF of 5, as assumed in this study for the evaluation of PS migration, following the assessment of EFSA for PET [5], are questionable. Individual OEFs for each applied surrogate are more suitable for the evaluation of the migration of post-consumer substances from recyclates based on diffusion modelling.

**Funding:** This research received no external funding.

**Data Availability Statement:** Not applicable.

**Acknowledgments:** This study was financed by Fraunhofer IVV own resources (internal project 509426). Thanks are due to Silvia Demiani, Johann Ewender, Anita Gruner, and Norbert Rodler for experimental contributions to this work. Special thanks are due to Frank Eisenträger (Ineos Styrolution), Ken Huestebeck, Jens Kathmann (both Styrenics Circular Solutions) and Mark Pemberton (Systox) for fruitful discussions and proof-reading of the manuscript.

**Conflicts of Interest:** The author declares no conflict of interest.

**Disclaimer:** The discussion and safety assessment of the recycling of post-consumer PS recyclates into new packaging in contact with food is the personal science-based opinion of the author, and it is not guaranteed that EFSA would evaluate these applications in the same way and with the same conclusions.

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
