# Peer review of "Recycling of Post-Consumer Polystyrene Packaging Waste into New Food Packaging Applications—Part 1: Direct Food Contact"

_recycling, doi:10.3390/recycling8010026_

Round 1

Reviewer 1 Report

The article provides useful insight on the suitability of PS recyclates for food contact use. There are no official criteria available for evaluating the PS recyclates, therefore the author performs a comparative assessment with existing evaluation criteria for PET and HDPE. Although no strong conclusive results are possible, the author provides a fair and empirically supported estimation. Therefore, I believe this article is useful for future assessments and the establishments of concrete criteria. I recommend this article for publication with a few minor revisions.

Here are my suggestions for revising this article:

1. In the abstract, lines 8 and 16 need revision. Not clear sentences, grammar mistakes.

2. In general, thorough proofreading is needed on the whole text, and especially in the Introduction section.

3. Please replace the work “Recollection” with the word “collection” throughout the text. The process of separately collecting plastic waste for recycling is called in literature “separate collection”. The word “recollection” has a different meaning in the English language.

4. In line 144, the aim of the study “is”, not “was”.

5. In lines 153-56, the disclaimer should be moved to other section. Probably in method, or conclusions presenting the limitations of the study. But not in Introduction.

6. In Figure 1, the names and numbers in parenthesis are confusing, as it appears identical in all 4 graphs. Please remove, or explain what these numbers mean in the caption of Figure 1.

Author Response

The article provides useful insight on the suitability of PS recyclates for food contact use. There are no official criteria available for evaluating the PS recyclates, therefore the author performs a comparative assessment with existing evaluation criteria for PET and HDPE. Although no strong conclusive results are possible, the author provides a fair and empirically supported estimation. Therefore, I believe this article is useful for future assessments and the establishments of concrete criteria. I recommend this article for publication with a few minor revisions.

Here are my suggestions for revising this article:

  1. In the abstract, lines 8 and 16 need revision. Not clear sentences, grammar mistakes.

ANSWER OF THE AUTHOR: the sentences are changed.

  1. In general, thorough proofreading is needed on the whole text, and especially in the Introduction section.

ANSWER OF THE AUTHOR: I revised the manuscript thoroughly.

  1. Please replace the work “Recollection” with the word “collection” throughout the text. The process of separately collecting plastic waste for recycling is called in literature “separate collection”. The word “recollection” has a different meaning in the English language.

ANSWER OF THE AUTHOR: Recollection was changed to collection in the whole manuscript.

  1. In line 144, the aim of the study “is”, not “was”.

ANSWER OF THE AUTHOR: Your suggestion has been accepted.

  1. In lines 153-56, the disclaimer should be moved to other section. Probably in method, or conclusions presenting the limitations of the study. But not in Introduction.

ANSWER OF THE AUTHOR: I moved this disclaimer to the end of the manuscript.

  1. In Figure 1, the names and numbers in parenthesis are confusing, as it appears identical in all 4 graphs. Please remove, or explain what these numbers mean in the caption of Figure 1.

ANSWER OF THE AUTHOR: I removed the concentrations from the graphs.

Reviewer 2 Report

In this paper the author uses the existing EFSA evaluation criteria for PET to assess the safety of post-consumer recycled PS for food contact applications.

The topic is of crucial interest considering the environmental issues related to plastic waste and the European initiatives for the transition towards a circular economy of plastics. Moreover, a systematic analysis of the state-of-art on PS recycling, with a focus on the actual information gaps for a thorough application of EFSA evaluation criteria to assess the safety of post-consumer PS for food contact applications, was conducted in this study.

In the referee’s opinion the paper deserves to be published in “Recycling” journal, but after some revisions. In the following, specific remarks are reported.

·    A careful reading of the whole manuscript is recommended in order to correct some typos and to improve the English syntactic construction of some sentences.

·     In the referee’s opinion the title of the paper should be modified in order to better recall the main topic of the study that is the application of EFSA safety criteria, established for PET, also for post-consumer recycled PS.

·    It is unclear if the surrogates, used for the challenge tests, were introduced inside the virgin HIPS or the recycled one. In the Introduction paragraph (lines 111- 114) the author writes “Within a challenge test, the post-consumer recyclates are artificially contaminated with model contaminants (surrogates)”, but in the paragraph §4.1 it is not specified if a recycled PS is used. The author should clarify this aspect.

·     At page 9 line 400 and page 10 line 10, the author refers to Figure 1, but it should be Figure 2. Please, correct. Moreover, in Figure 2 it should be better to report the points corresponding to the Minimum cleaning efficiencies and Molecular weights of the surrogates (also indicating the name of the surrogates for clarity) instead of the curves.

·     At page 9 lines 389-391 the author writes “As an example, the migration of the surrogate toluene (molecular weight: 92 g/mol) with a starting concentration of 3 mg/kg was calculated to 1.38 μg/kg in yogurt after storage for 40 days at 6 °C (storage conditions of cold filled yogurt)”: it is unclear how the author calculates the migration value “1.38 μg/kg”. Does the author apply the diffusion model? Please, specify better in the manuscript.

·   The over-estimated factors of the model migration predicted values compared to the experimental ones are very high for some surrogates, specifically at the lower temperatures investigated: maybe it would more suitable to measure the specific migration values using the experimental procedure established within the Regulation EU 2016/1416.

·    The paragraph with the Conclusions is too long: the author should just report the more relevant results of this study and the points that need further inspections.

Author Response

In this paper the author uses the existing EFSA evaluation criteria for PET to assess the safety of post-consumer recycled PS for food contact applications.

The topic is of crucial interest considering the environmental issues related to plastic waste and the European initiatives for the transition towards a circular economy of plastics. Moreover, a systematic analysis of the state-of-art on PS recycling, with a focus on the actual information gaps for a thorough application of EFSA evaluation criteria to assess the safety of post-consumer PS for food contact applications, was conducted in this study.

In the referee’s opinion the paper deserves to be published in “Recycling” journal, but after some revisions. In the following, specific remarks are reported.

  • A careful reading of the whole manuscript is recommended in order to correct some typos and to improve the English syntactic construction of some sentences.

ANSWER OF THE AUTHOR: I revised the manuscript thoroughly.

  • In the referee’s opinion the title of the paper should be modified in order to better recall the main topic of the study that is the application of EFSA safety criteria, established for PET, also for post-consumer recycled PS.

ANSWER OF THE AUTHOR: Thanks for this comment. I agree, that in this manuscript the focus is about the transfer of the evaluation of the criteria for rPET to rPS. The change of title possibly makes sense. But it will be a series of publications (most probably three). Only this manuscript deals with the EFSA criteria. I have therefore decided not to follow your recommendation and not to change the title

  • It is unclear if the surrogates, used for the challenge tests, were introduced inside the virgin HIPS or the recycled one. In the Introduction paragraph (lines 111- 114) the author writes “Within a challenge test, the post-consumer recyclates are artificially contaminated with model contaminants (surrogates)”, but in the paragraph §4.1 it is not specified if a recycled PS is used. The author should clarify this aspect.

ANSWER OF THE AUTHOR: This is indeed correct: I added some sentences into the manuscript, which descripted more clearly what we have done.

  • At page 9 line 400 and page 10 line 10, the author refers to Figure 1, but it should be Figure 2. Please, correct. Moreover, in Figure 2 it should be better to report the points corresponding to the Minimum cleaning efficiencies and Molecular weights of the surrogates (also indicating the name of the surrogates for clarity) instead of the curves.

ANSWER OF THE AUTHOR: Figure 1 was corrected to Figure 2. In addition I introduced a revised Figure 2 into manuscript with dashed lines for the applied surrogates.

  • At page 9 lines 389-391 the author writes “As an example, the migration of the surrogate toluene (molecular weight: 92 g/mol) with a starting concentration of 3 mg/kg was calculated to 1.38 μg/kg in yogurt after storage for 40 days at 6 °C (storage conditions of cold filled yogurt)”: it is unclear how the author calculates the migration value “1.38 μg/kg”. Does the author apply the diffusion model? Please, specify better in the manuscript.

ANSWER OF THE AUTHOR: Yes , calculation were done with the AP prediction model. I included this into the manuscript.

  • The over-estimated factors of the model migration predicted values compared to the experimental ones are very high for some surrogates, specifically at the lower temperatures investigated: maybe it would more suitable to measure the specific migration values using the experimental procedure established within the Regulation EU 2016/1416.

ANSWER OF THE AUTHOR: Thanks for this valuable comment. Measuring the migration into simulants is also over-estimative, because the simulants significantly swell the PS matrix. I included a two explaining sentences into the manuscript and added a reference.

  • The paragraph with the Conclusions is too long: the author should just report the more relevant results of this study and the points that need further inspections.

ANSWER OF THE AUTHOR: The conclusions section of the manuscript is around one page, which is in my point of view not too long compared to the rest of the sections. However, I shortened the conclusion section a little bit.

Reviewer 3 Report

Dear Author,
First of all, I have some general remarks:
You always use to put a comma before "that" (".. this shows, that..", "... due to the fact, that..."), which is not common in English ("...this shows that the approach is wrong...") and at least irritates when reading, sometimes even changes the meaning. Please correct this throughout the text.
Furthermore, due to the fact that the chapter "Materials and Methods" comes after the discussion, it is not clear until then how to arrive at the calculated migration values. Thus, the reader has difficulties to distinguish between measured values and values derived from the models. Here, one should refer more often in the text, especially in chapters 2 and 3, to the corresponding section in chapter 4 (especially when talking about "predicted migration") and also be very careful in the choice of words. Indeed, it is not always clear in the text whether it is precisely a maximum permissible value, a value measured in real terms, or an estimated value that is meant.
In addition, the term "migration" primarily stands for the transport process itself. If one wants to refer to a concrete migrated amount of substance, one should speak of a "migrated amount" or "migrated quantity" (e.g. in Table 1). It has been done differently for decades and the amount of substance is also referred to as "migration", but this does not make it easier for the normal reader to understand.
The following other terms are used more frequently, are not readily understood, and should be clearly defined in the text when first mentioned:
- Consumer exposure
- Packaging concentration (more precisely: concentration of a contaminating substance in the packaging)
- Post-consumer substance (more precisely: substance that is introduced into the material after use of the package)
The frequently used word "recollection" is ambiguous and should be replaced by "collection", where it fits also by "curbside collection". In some cases (see individual comments), it is also appropriate to replace "recollected" with "recovered".
Individual comments:
Line 8: "is" instead of "as".
Line 15: "comparable" is wrong. You can always compare two things. "similar" would fit, for example.
Line 52: PET collection systems
Line 63: ... are collected in curbside systems (e.g. like the "green dot") ...
Line 70: ... conservative evaluation criteria ...
Line 74: ...has not been established ...
Line 79: delete HIPS and GPPS (double)
Line 87: I would rather say "excluded" instead of "discounted".
Line 102: Say more precisely what is meant by "input stream".
Line 106: "EFSA assumed that all of ..."
Lines 114 / 115: The sentence "The concentrations ... material." Is unclear. Be more specific about what concentrations are meant.
Line 116: Is it stated somewhere in which ratio the contaminated material was added to the other?
Line 120: What are "out materials"? (See also line 102: "input stream")
Line 136: "Recovered PS from post-consumer sources is available..."
Line 144: "determine" instead of "discover".
Line 158: collection
Line 162: the sentence is misleading. It is thought that all bales contain mostly PS items, which cannot be so and is probably not what is meant
Line 164: "expanded PS" is not a "non-PS".
Line 166: ... articles are ...
Line 170: I would call cups "semi-rigid", so: ... rigid and semi-rigid ...
Line 175 to 177: Revise sentence
Line 178: Color sorting is done with visible light!
Line 179: ... in a white and colored fraction of PS flakes, each of 99.8% purity.
Line 180: ...PS flakes need ...
Line 188: ...steps had been used ...
Line 189: Who is meant by "they"?
Line 197: With regard to ... or ...Regarding the ...
Line 199: What is meant: ... might be a reason for ... or ... might be a consequence of the fact ...?
Line 201: Therefore, only a limited amount of post-consumer PS samples could be analyzed with regard to ...
Line 202: ... substances like limonene were found in the range ...
Line 204: It should be noted that this study was conducted about 25 years ago.
Line 207: ...substances...
Line 211: ... PS containers... (it's not the materials that you close).
Line 212: ...chemically resistant
Lines 216 - 218: sentence difficult to understand
Lines 221 - 223: What does the first half sentence logically have to do with the second?
Line 231: The specific migration of the different substances at 60 °C...
Line 233: Integrate reference to method section in table heading
Table 1: migrated amount
Line 236: Note in figure caption: Spiking levels for the different substances are indicated in the figure legends.
Line 238: This is section 2.3! Again refer to method section
Line 246: ... an exposure of 0.0025 µg of a substance per kg of body mass at maximum is not critical ...
Line 247: The exposure-related amount (or limit) for a substance... (it is not the actual exposure, but the limit value)
Lines 248 / 249: This limit ... can be considered as the maximum admissible value for the migrated quantity (as it is written, it could also be considered as the maximum actually migrated quantity)
Line 257: The consumption data for drinks...
Line 266: ... the lower part of the packaging...
Line 269: ... as a minimum
Lines 271 / 272: Migration scenario for calculations for toddlers with 10 kg body weight. Values for food consumption, maximum storage temperatures, admissible exposure and values for maximum migrated quantity are indicated.
Line 308: What kind of diffusion is meant here? Of the individual packages in the collection?
Lines 317 to 319: This sentence should be worded differently: In the usual curbside collection, larger non-food EPS packagings are being collected and also insulation materials are often improperly disposed of. This contaminates the entire EPS stream. Here, therefore, the requirement must be placed on the sorting systems that EPS objects do not end up in the recycling path for PS containers.  Therefore, the sentence should rather read: Provided that EPS and PS objects are separated in the sorting process and EPS objects do not enter the PS recycling stream considered here, contamination of the PS recyclates by non-food PS is unlikely.
Line 335: again indicate the method of calculation
Line 338: ... are smaller than...
Lines 343 to 345: Reference to method part, so that one can understand it
Line 362: ...it should be made sure that the exposure...
Line 368: ...lower than predicted...
Line 378: ...maximum admissible consumer exposure...
Line 387: ...the maximum migrated quantity into packed food
Lines 400 / 401: Figure 2!
Line 406: ... compared to the...
Line 410: ... higher than...
Line 424: ... in zero required cleaning efficiencies...
Line 453: ... dichloromethane at 40 °C for 3 days.
Line 454: ... in order to determine ...
Line 460: If acetone and ethyl acetate were not determined, why mention them?
Line 467: Quantities migrated into...
Line 489: ... 100% for the food contact article containing recyclate...
Line 497: As a logical consequence,...
Lines 525 to 527: The most favorable applications for post-consumer recyclates are foods for refrigerated storage conditions such as cold filled yogurts or meat, fish or cheese.
Lines 547/548: This opens the way for experimental migration testing also to be used for safety assessment of PS recyclates.

Author Response

First of all, I have some general remarks:

You always use to put a comma before "that" (".. this shows, that..", "... due to the fact, that..."), which is not common in English ("...this shows that the approach is wrong...") and at least irritates when reading, sometimes even changes the meaning. Please correct this throughout the text.

ANSWER OF THE AUTHOR: Thanks for this recommendation. Indeed I'm not a native speaker. I went through the whole document and deleted the commas before "that"

Furthermore, due to the fact that the chapter "Materials and Methods" comes after the discussion, it is not clear until then how to arrive at the calculated migration values. Thus, the reader has difficulties to distinguish between measured values and values derived from the models. Here, one should refer more often in the text, especially in chapters 2 and 3, to the corresponding section in chapter 4 (especially when talking about "predicted migration") and also be very careful in the choice of words. Indeed, it is not always clear in the text whether it is precisely a maximum permissible value, a value measured in real terms, or an estimated value that is meant.

ANSWER OF THE AUTHOR: Thanks for this comment. It is a requirement of the journals that the chapter is placed after the discussion chapter. I personally agree with you. However, I made no changes in the manuscript, because it is part of the "Recycling" template.

In addition, the term "migration" primarily stands for the transport process itself. If one wants to refer to a concrete migrated amount of substance, one should speak of a "migrated amount" or "migrated quantity" (e.g. in Table 1). It has been done differently for decades and the amount of substance is also referred to as "migration", but this does not make it easier for the normal reader to understand.

ANSWER OF THE AUTHOR: I went through the whole document and changes "migration" into "migrated amount" where applicable

The following other terms are used more frequently, are not readily understood, and should be clearly defined in the text when first mentioned:

- Consumer exposure

ANSWER OF THE AUTHOR: I changed this in the whole document

- Packaging concentration (more precisely: concentration of a contaminating substance in the packaging)

ANSWER OF THE AUTHOR: Packaging concentration was found two times in the manuscript. Both section were changed according to your recommendations.

- Post-consumer substance (more precisely: substance that is introduced into the material after use of the package)

ANSWER OF THE AUTHOR: I introduced the explanation into section 2.2

The frequently used word "recollection" is ambiguous and should be replaced by "collection", where it fits also by "curbside collection". In some cases (see individual comments), it is also appropriate to replace "recollected" with "recovered".

ANSWER OF THE AUTHOR: I change recollection to collection in the whole document

Individual comments:

Line 8: "is" instead of "as".

ANSWER OF THE AUTHOR: Your suggestion has been accepted.

Line 15: "comparable" is wrong. You can always compare two things. "similar" would fit, for example.

ANSWER OF THE AUTHOR: Your suggestion has been accepted.

Line 52: PET collection systems

ANSWER OF THE AUTHOR: Your suggestion has been accepted.

Line 63: ... are collected in curbside systems (e.g. like the "green dot") ...

ANSWER OF THE AUTHOR: Your suggestion has been accepted.

Line 70: ... conservative evaluation criteria ...

ANSWER OF THE AUTHOR: Your suggestion has been accepted.

Line 74: ...has not been established ...

ANSWER OF THE AUTHOR: Your suggestion has been accepted.

Line 79: delete HIPS and GPPS (double)

ANSWER OF THE AUTHOR: Your suggestion has been accepted.

Line 87: I would rather say "excluded" instead of "discounted".

ANSWER OF THE AUTHOR: Your suggestion has been accepted.

Line 102: Say more precisely what is meant by "input stream".

ANSWER OF THE AUTHOR: I added "materials for the recycling process"

Line 106: "EFSA assumed that all of ..."

ANSWER OF THE AUTHOR: Your suggestion has been accepted.

Lines 114 / 115: The sentence "The concentrations ... material." Is unclear. Be more specific about what concentrations are meant.

ANSWER OF THE AUTHOR: I added "which are applied in a challenge test"

Line 116: Is it stated somewhere in which ratio the contaminated material was added to the other?

ANSWER OF THE AUTHOR: " without dilution with non-contaminated material" added.

Line 120: What are "out materials"? (See also line 102: "input stream")

ANSWER OF THE AUTHOR: out should be read as ouptput. Corrected.

Line 136: "Recovered PS from post-consumer sources is available..."

ANSWER OF THE AUTHOR: Your suggestion has been accepted.

Line 144: "determine" instead of "discover".

ANSWER OF THE AUTHOR: Your suggestion has been accepted.

Line 158: collection

ANSWER OF THE AUTHOR: Your suggestion has been accepted.

Line 162: the sentence is misleading. It is thought that all bales contain mostly PS items, which cannot be so and is probably not what is meant

ANSWER OF THE AUTHOR: The bales contain presorted waste when entering the PS recycling plant, so mainly PS containers entered the plants. The sentence is in my point of view not misleading.

Line 164: "expanded PS" is not a "non-PS".

ANSWER OF THE AUTHOR: expanded PS deleted.

Line 166: ... articles are ...

ANSWER OF THE AUTHOR: Your suggestion has been accepted.

Line 170: I would call cups "semi-rigid", so: ... rigid and semi-rigid ...

ANSWER OF THE AUTHOR: Your suggestion has been accepted.

Line 175 to 177: Revise sentence

ANSWER OF THE AUTHOR: Sentence was revised.

Line 178: Color sorting is done with visible light!

ANSWER OF THE AUTHOR: Correct. Sentence changed.

Line 179: ... in a white and colored fraction of PS flakes, each of 99.8% purity.

ANSWER OF THE AUTHOR: Your suggestion has been accepted.

Line 180: ...PS flakes need ...

ANSWER OF THE AUTHOR: Your suggestion has been accepted.

Line 188: ...steps had been used ...

ANSWER OF THE AUTHOR: Your suggestion has been accepted.

Line 189: Who is meant by "they"?

ANSWER OF THE AUTHOR: PS super-clean recycling processes included instead of "they".

Line 197: With regard to ... or ...Regarding the ...

ANSWER OF THE AUTHOR: Your suggestion has been accepted.

Line 199: What is meant: ... might be a reason for ... or ... might be a consequence of the fact ...?

ANSWER OF THE AUTHOR: I deleted the second part if the sentence.

Line 201: Therefore, only a limited amount of post-consumer PS samples could be analyzed with regard to ...

ANSWER OF THE AUTHOR: Your suggestion has been accepted.

Line 202: ... substances like limonene were found in the range ...

ANSWER OF THE AUTHOR: Your suggestion has been accepted.

Line 204: It should be noted that this study was conducted about 25 years ago.

ANSWER OF THE AUTHOR: Your suggestion has been accepted.

Line 207: ...substances...

ANSWER OF THE AUTHOR: Your suggestion has been accepted.

Line 211: ... PS containers... (it's not the materials that you close).

ANSWER OF THE AUTHOR: Your suggestion has been accepted.

Line 212: ...chemically resistant

ANSWER OF THE AUTHOR: Your suggestion has been accepted.

Lines 216 - 218: sentence difficult to understand

ANSWER OF THE AUTHOR: The sentence was changed.

Lines 221 - 223: What does the first half sentence logically have to do with the second?

ANSWER OF THE AUTHOR: Due to the fact, that post-consumer substances are determined in PS only in low concentrations, they cannot be used as migrants for kinetics. Therefore artificial contamination was applied with much higher concentrations. I added "in high concentrations" to make the sentence clearer.

Line 231: The specific migration of the different substances at 60 °C...

ANSWER OF THE AUTHOR: Your suggestion has been accepted.

Line 233: Integrate reference to method section in table heading

ANSWER OF THE AUTHOR: The method is described in section 4.3, but I think this is not necessary to mention in the Table header.

Table 1: migrated amount

ANSWER OF THE AUTHOR: Your suggestion has been accepted.

Line 236: Note in figure caption: Spiking levels for the different substances are indicated in the figure legends.

ANSWER OF THE AUTHOR: Spiking levels were removed from the figure legends. Reference to Table 5 included instead.

Line 238: This is section 2.3! Again refer to method section

ANSWER OF THE AUTHOR: Section number changed

Line 246: ... an exposure of 0.0025 µg of a substance per kg of body mass at maximum is not critical ...

ANSWER OF THE AUTHOR: Your suggestion has been accepted.

Line 247: The exposure-related amount (or limit) for a substance... (it is not the actual exposure, but the limit value)

ANSWER OF THE AUTHOR: Your suggestion has been accepted.

Lines 248 / 249: This limit ... can be considered as the maximum admissible value for the migrated quantity (as it is written, it could also be considered as the maximum actually migrated quantity)

ANSWER OF THE AUTHOR: Your suggestion has been accepted.

Line 257: The consumption data for drinks...

ANSWER OF THE AUTHOR: Your suggestion has been accepted.

Line 266: ... the lower part of the packaging...

ANSWER OF THE AUTHOR: Your suggestion has been accepted.

Line 269: ... as a minimum

ANSWER OF THE AUTHOR: Your suggestion has been accepted.

Lines 271 / 272: Migration scenario for calculations for toddlers with 10 kg body weight. Values for food consumption, maximum storage temperatures, admissible exposure and values for maximum migrated quantity are indicated.

ANSWER OF THE AUTHOR: Your suggestion has been accepted.

Line 308: What kind of diffusion is meant here? Of the individual packages in the collection?

ANSWER OF THE AUTHOR: Diffusion behavior in general. PET and PS are low diffusive polymers. The sentence was not correct. I exchanged "between" with "of".

Lines 317 to 319: This sentence should be worded differently: In the usual curbside collection, larger non-food EPS packagings are being collected and also insulation materials are often improperly disposed of. This contaminates the entire EPS stream. Here, therefore, the requirement must be placed on the sorting systems that EPS objects do not end up in the recycling path for PS containers.  Therefore, the sentence should rather read: Provided that EPS and PS objects are separated in the sorting process and EPS objects do not enter the PS recycling stream considered here, contamination of the PS recyclates by non-food PS is unlikely.

ANSWER OF THE AUTHOR: Your suggestion has been accepted. However with a slightly different sentence.

Line 335: again indicate the method of calculation

ANSWER OF THE AUTHOR: The calculations were done by use of the AP model as described in the experimental section. The AP model is already meantion in this sentence. Therefore I haven't changed the sentence.

Line 338: ... are smaller than...

ANSWER OF THE AUTHOR: reference to section 4.4 included.

Lines 343 to 345: Reference to method part, so that one can understand it

ANSWER OF THE AUTHOR: reference to section 4.4 included.

Line 362: ...it should be made sure that the exposure...

ANSWER OF THE AUTHOR: Your suggestion has been accepted.

Line 368: ...lower than predicted...

ANSWER OF THE AUTHOR: Your suggestion has been accepted.

Line 378: ...maximum admissible consumer exposure...

ANSWER OF THE AUTHOR: Your suggestion has been accepted.

Line 387: ...the maximum migrated quantity into packed food

ANSWER OF THE AUTHOR: Your suggestion has been accepted.

Lines 400 / 401: Figure 2!

ANSWER OF THE AUTHOR: Corrected

Line 406: ... compared to the...

ANSWER OF THE AUTHOR: Your suggestion has been accepted.

Line 410: ... higher than...

ANSWER OF THE AUTHOR: Your suggestion has been accepted.

Line 424: ... in zero required cleaning efficiencies...

ANSWER OF THE AUTHOR: Your suggestion has been accepted.

Line 453: ... dichloromethane at 40 °C for 3 days.

ANSWER OF THE AUTHOR: Your suggestion has been accepted.

Line 454: ... in order to determine ...

ANSWER OF THE AUTHOR: Your suggestion has been accepted.

Line 460: If acetone and ethyl acetate were not determined, why mention them?

ANSWER OF THE AUTHOR: I deleted the sentence

Line 467: Quantities migrated into...

ANSWER OF THE AUTHOR: Your suggestion has been accepted.

Line 489: ... 100% for the food contact article containing recyclate...

ANSWER OF THE AUTHOR: Here I did not follow your suggestion, but I deleted "recyclate containing" in the sentence because recyclate was mentioned twice.

Line 497: As a logical consequence,...

ANSWER OF THE AUTHOR: Your suggestion has been accepted.

Lines 525 to 527: The most favorable applications for post-consumer recyclates are foods for refrigerated storage conditions such as cold filled yogurts or meat, fish or cheese.

ANSWER OF THE AUTHOR: Your suggestion has been accepted.

Lines 547/548: This opens the way for experimental migration testing also to be used for safety assessment of PS recyclates.

ANSWER OF THE AUTHOR: Your suggestion has been accepted.